# Micro-Electro-Mechanical Systems in Light Stabilization

**DOI:** 10.3390/s23062916

**Published:** 2023-03-08

**Authors:** Marian Gilewski

**Affiliations:** Faculty of Electrical Engineering, Bialystok University of Technology, 45A Wiejska Street, 15-351 Bialystok, Poland; m.gilewski@pb.edu.pl

**Keywords:** MEMS-spectrometer, signal conditioning, FPGA in control circuits

## Abstract

This article discusses application considerations in the micro-electro-mechanical system’s optical sensor. Furthermore, the provided analysis is limited to application issues occurring in research or industrial applications. In particular, a case was discussed where the sensor was used as a feedback signal source. Its output signal is used to stabilize the flux of an LED lamp. Thus, the function of the sensor was the periodic measurement of the spectral flux distribution. The application problem of such a sensor is the output analogue signal conditioning. This is necessary to perform analogue-to-digital conversion and further digital processing. In the discussed case, design limitations come from the specifics of the output signal. This signal is a sequence of rectangular pulses, which can have different frequencies, and their amplitude varies over a wide range. The fact such a signal must be conditioned additionally discourages some optical researchers from using such sensors. The developed driver allows measurement using an optical light sensor in the band from 340 nm to 780 nm with a resolution of about 12 nm; in the range of flux values from about 10 nW to 1 μW, and frequencies up to several kHz. The proposed sensor driver was developed and tested. Measurement results are presented in the paper’s final part.

## 1. Introduction

Sensor miniaturization, distributed measurement systems, and complex measurement data processing algorithms are becoming inherent functionalities of modern measurement systems. This trend is also observed in optical metrology [1,2] and it is resulting in new application perspectives [3,4,5]. Applications of new technologies using integrated optical modules are also increasing in spectral metrology. Conventional benchtop spectrometers typically consist of an input collimator system, a multi-stage controlled monochromator, a sensitive photodetector, and a desktop computer for system control and data processing [6,7,8]. The stationary nature of these instruments made it possible to achieve high spectral resolutions, obtain a wide measurement bandwidth, and support large dynamic ranges of the input signal. There were no design space or energy constraints in these solutions. In such measurement systems, in addition to complex electronics, there are specialized mechanical systems that drive the monochromator modules. Consequently, some [9] of such designs consume power of several thousand watts, they weigh several hundred kilograms, and occupy a space of several cubic meters. The high power consumption and large size of these instruments, practically eliminate them from mobile applications, including control circuits of measurement systems.

Technological developments in monochromator miniaturization and their integration with CCD arrays have resulted in the construction of a new spectrometer category, namely, mini-spectrometers. The complete measurement system of the mini-spectrometer consists of two basic modules: a monochromator integrated with a CCD and an external portable laptop computer. Since the monochromator module is in volume about 1 cubic decimeter and consumes several watts in power, the whole system is more mobile than a stationary spectrometer. An important advantage of these systems is the elimination of mechanically driven multi-stage monochromators. The monochromator module typically consists of an entrance slit, collimating lens, transmission or reflecting grating, focusing mirror, linear CCD sensor, and required electronics (Figure 1).

Similar to the use of other technologies, mini-spectrometers have a certain level of measurement errors. In this case, the sources of errors are dark current, thermal sensitivity, stray light, photoelectric conversion nonlinearity, and electronic circuit errors. In the available bibliography, you can find descriptions of methods, techniques, and tools for minimizing the impact of these error sources [11]. Measuring instruments in CCD spectrometers technology, due to their mobility, have found their place in laboratory instruments and they are partially replacing stationary spectrometers.

However, the progress of miniaturization has not stopped at the mini-spectrometer level, because optical sensors using micro-electro-mechanical systems (MEMS) technology have been developed. In the meantime, a number of scientific publications have been produced on research in various areas of these sensors’ applications [12,13,14,15,16,17,18,19,20]. Unfortunately, there have also been critical opinions about the MEMS technology applications in spectral analysis. This state of affairs has reinforced the concerns of some optical researchers and resulted in a distrustful approach to these chips’ applications in their laboratories.

The primary objection to optical MEMS spectrometers is the high level of stray light. Of course, some level of stray light must occur in a MEMS structure. However, it does not differ in value compared to other spectrometers containing single-stage monochromators. In some MEMS spectrometer solutions [21], stray light attenuation is about −30 dB. This is the stray light attenuation value that is achieved in CCD mini-spectrometers, which are considered mid-range measurement instruments. The second criticized parameter in MEMS spectrometers is the value of spectral resolution. In fact, in the visible range, this is almost twice the value possible for mini spectrometers. The purpose of MEMS spectrometers is not to replace laboratory measurement equipment. Therefore, a slightly higher value of spectral resolution is not a critical parameter in this case. This value is sufficient for many industrial, scientific, medical, or agricultural applications [22].

The topic of this paper is not the analysis of the optical path in the MEMS spectrometer. This issue has been studied and published in the scientific literature, in which many photoelectric conversion solutions are presented. This paper is concerned with applied research and not fundamental research, hence the used writing style of the paper is similar to reports of such research rather than a comprehensive analytical approach. The author of the paper wishes to pay attention to the application issues of MEMS spectrometers, which significantly affect the qualitative parameters of the whole system. The technical documentation supplied by the MEMS spectrometer manufacturer is limited, and it does not suggest specific application solutions and their possible versions. Such a state may discourage many potential users of these sensors, hence the author of this paper proposes his own original solution for the MEMS driver, which is his contribution to the scope of applied research on the subject. The purpose of this paper is to pay attention to application limitations and to present proposals for low-complexity MEMS spectrometer support systems.

## 2. Materials and Methods

This experiment was carried out using the C12666MA [23] MEMS spectrometer with dispersive optics. Its internal structure is shown in Figure 2. This module is housed in a metal case with dimensions: 20.1 mm × 12.5 mm × 10.1 mm and it weighs about 5 g. All the electronic signal pins are of 10-pin DIP standard. 

This chip was used in the greenhouse lamp hardware simulator. Because LED lamps have unstable optical parameters in time, it is necessary to correct their flux. For this purpose, it is required to monitor the flux value and its spectral distribution online. However, in the described application, very precise spectral resolution is not required, but a wide dynamic range of measured flux values is. For this purpose, the MEMS spectrometer is an adequate solution for the measurement path. A stand-alone MEMS sensor does not become a complete measurement system; it requires a power supply and handling of the control signals and measurement output signal. 

In the described example, the complete simulator system consisted of 4 basic modules, shown in Figure 3. The original, proprietary LED matrix solution (Figure 4a) is made up of 25 LEDs with different spectral distributions, which are powered independently. The set of current values for each LED determines the resulting spectral distribution of the lamp. These currents are determined by the power matrix and cooling module, in such a way as to obtain the spectral characteristics of the source. This characteristic should be similar to the absorption curves of plants [24], which have been developed through basic research [25,26,27]. All current settings of this module can be adjusted automatically with the FPGA module [28] or manually. The FPGA module is only the hardware layer, for which it is necessary to develop appropriate software. An FPGA chip was chosen for the project instead of a microcontroller because it allows multiple control processes to be performed independently of each other and at the same time. The current LED array flux was measured by the C1266MA chip (Figure 4b) and then converted to digital signals in the MEMS spectrometer module. The FPGA module compares the measured flux distribution of the LED array with the reference distribution stored in its memory. If these values differ significantly, the FPGA module sends a command to the supply matrix and cooling module to decrease or increase the currents supplying particular LEDs. 

The described system of adaptive and reference LED sources is complex and many-sided, so it has not been described in detail in this paper. This paper deals with only one component of this system, but the introductory background had to be described.

All the ancillary components of the MEMS sensor, such as voltage regulators, isolation amplifier, A/D converter, and level translators, were mounted on the MEMS circuit prototype board. This board was fixed in the integrating sphere entrance (Figure 5), in the way that the MEMS sensor was oriented to the inside of the labsphere. The LED matrix was mounted in the second entrance of the integrating sphere, directing the active side of the matrix to the sphere’s cavity also. All driver modules were connected to each other and to the power supply and the oscilloscope in the way shown in Figure 5. In this figure, for better visibility, the cooling and thermal stabilization modules of the LED array are not shown. These are the Peltier module and the Master Liquid Lite 240 active cooler. Both the Peltier module and the active cooler head are mounted on the external side of the LED matrix normally. These modules are powered by the matrix control board and can be controlled by the FPGA board. 

The mentioned measurement bench (Figure 5) will finally be used to build the prototype hardware simulator of the horticulture lamp. On the other hand, at the early stage of the research it was used to measure and evaluate the applicability of the MEMS sensor in the whole simulator system. At the same time, it was the base for conducting studies of the MEMS driver proposed and described in this paper. Hence, the view of the test bench in Figure 5 is not compatible with the overall block diagram of the lamp simulator in Figure 3.

In an early development, the MEMS circuit board was built according to the schematic shown in Figure 6, which is recommended by the sensor manufacturer. This scheme includes the following: a U1—MEMS spectrometer, U2—amplifier/separator, U4—A/D converter, and U3 and U5–U7 logic level translators. The U2 unit-gain amplifier separates the C12666MA output from the rest of the circuit. The analogue signal from the separating amplifier is converted into a digital signal by the U4 converter. The digital signal from the U4 converter is transmitted via the SPI bus, which contains the Dout, CS, and SCLK signals. Since all the digital signals in the described circuit are in the 5 V standard, it is necessary to convert them to the 3.3 V standard, which is the dominant standard in FPGA chips and microcontrollers. In the presented circuit, the conversion of the digital signals between the 3.3 V and 5 V standards is performed by chips U3, U5, U6, and U7.

The most important time characteristics of the sensor were investigated in the above system, and the measurement results indicated the disadvantages of the default circuit. These results have been the motivation to develop the new MEMS spectrometer driver, which is described below.

## 3. Results and Discussion

The investigated MEMS sensor has a non-linear spectral response characteristic (Figure 7). However, this is not a problem, as its correction can be realized analytically in the FPGA, or by hardware shortening or lengthening the integration times in individual spectral intervals. This property is not present in most stationary spectrometers or mini-spectrometers, where the integration time is fixed.

Similarly, by changing the integration time or gain of the measurement path, the non-linearity of the spectral resolution MEMS spectrometer can be corrected (Figure 8). 

The MEMS C12666CA is also characterized by non-linearity in the photoelectric conversion (Figure 9), especially for short integration times. Hence, digital correction of the measured flux values on the FPGA side is required.

It is preferable to operate the sensor with the longest possible integration times due to the sensor’s relatively high inherent noise level of 0.5 mV. Since most of the noise is characterized by a zero mean value, their averaging is important to improve the signal-to-noise ratio at the system output. Thus, in addition to the choice of integration times, appropriate amplification by the sensor driver is also important. The appropriate development of the driver-amplifier architecture and parameters was the essence of the present work. The main disadvantage of the circuit shown in Figure 6 is the mismatch between the analogue part of the measurement path and the video signal characteristics at the output of the C12666MA sensor. The photodetection chip of the MEMS spectrometer allows the measurement of flux values over a wide range, from sub-pico watts to almost 1 microwatt. 

This means that in a sensor powered by +5 V, the output video signal should change between about 1 uV and a few V. Analogue-to-digital conversion of the input signal at the level of a few μV is practically impossible because the measured variable signal appears on a background of a couple of volts DC component. 

Such a high value of the DC component at the sensor output is misinterpreted by some optical metrologists as a component of the output signal caused by stray light. For this reason, some metrologists are abandoning the use of MEMS sensors. The DC component of a few volts cannot be produced by stray light, as it is internally attenuated against incoming light at −25 to −33 dB. This DC component results from the normal performance of an electronic circuit, especially when such a circuit is supplied with an unsymmetric voltage and the operating point of its output stage is near class A amplifiers. Figure 10 shows selected waveforms in the MEMS spectrometer circuit. The yellow waveform is the Video signal at the output of amplifier U2 shown in Figure 6. The U2 circuit was a low-voltage rail-to-rail input/output operational amplifier. Therefore, it did not significantly modify the Video signal from the output of the C12666MA chip. As shown in the figure, the video signal (yellow waveform) is a repeating sequence of rectangular pulses. It consists of approximately 40 pulses, each representing a spectral measurement interval of approximately 10 nm. The amplitude of the pulse corresponds to the flux measured in this interval. Hence, the amplitudes of all pulses represent the spectral distribution of the radiation over the entire visible range. Thus, the associated electronic circuit should measure the amplitudes of all pulses to reproduce the complete spectral distribution of the measured radiation. Due to the fact that only the variable component of the video signal provides information about the interval values of the streams—the DC component in the video signal is unnecessary. Therefore, from a measurement path efficiency point of view, the Video signal should be stripped of the DC component before performing the analogue-to-digital conversion, as shown in Figure 11. This is particularly important in the measurement of weak optical signals. This is because amplification of the integrated video signal, which contains both variable and DC components, would significantly reduce the dynamic range of the measurement path. 

Therefore, before analogue-to-digital conversion, the variable component and the DC component should be separated from the Video signal. Both these components need to be conditioned independently before entering the A/D converter. The following section presents the concept of the electronic circuit that performs the filtering functions of both components, amplifies the variable component, and matches its level to the input voltage range of the AD converter. The circuit shown in Figure 12, in addition to the C12666MA sensor and FPGA module and level translators, includes the following: an 8-input AD converter, the low-pass (LP) filter, and the high-pass (LP) filter together with the amplifier. 

From the above explanation, the methodology for conducting the research work was derived, in order:Development of the overall MEMS controller concept;Design of the electronic circuit according to the manufacturer’s scheme in Figure 6;Manufacturing of the controller board;FPGA software design;FPGA software validation;Measurements of the built system in the work bench in Figure 5;Critical analysis of the results obtained;Deciding on the development of a custom layout;New controller concept development;Assembly design of the new controller in Figure 9;Analytical calculations of the amplifier and filter circuit;Simulations of selected controller modules;Design implementation of the new controller;Modification of the FPGA software;Testing of the finished circuit on the work bench in Figure 5.

The output signals for the DC component from the LP output and the 7-channel output signal of the HP variable component flow to the inputs of the AD converter. The detailed implementation of the amplifier filter for the variable component is shown in Figure 13.

The input amplifier U1 separates the sensor output from the rest of the AC path. The integrated video signal from the output of U1 can be measured directly by the AD converter at the input of ADC1. At the same time, this signal flows through a second-order high-pass filter (parts: C3, R1, C4, R2, and 1/2 U2). This filter cuts out the DC component from the Video signal. The high-pass filter designed and used has a relatively low cut-off frequency of about 2 Hz (Figure 14). Its value, verified empirically, is due to the need to pass rectangular pulses as low as 200 Hz.

The filtered AC component is amplified in an 8-stage cascode amplifier. Part of the amplifier section connects directly to the AD converter inputs, viz: ADC2, ADC3, AC4, ADC5, ADC6, and ADC7. Each of the component stages amplifies the video pulses two times. This means that the A/D converter processes an AC signal that is amplified: once, twice, 8 times, 32 times, 64 times, and 128 times. Since the amplification values are multiples of two, this makes it easier to recalibrate the video signal values on the digital side because digital division by two means rejecting the LSB bit. The use of a multi-output amplifier, which is combined with a multi-input AD converter, avoids the construction of a complex single-stage amplifier. Depending on the converted amplitude of the video signal, the FPGA performs an automatic gain control function in real time. For this purpose, it chooses the signal from that input of the AD converter at which the video pulses are largest but still within the input range of the AD converter. The phase inversion of the video signal in the inverting amplifiers is not important, as the ADC is able to measure both the low and high levels of the video pulses and the FPGA can calculate the value of the level difference.

Figure 15 and Figure 16 show the waveforms of the amplified pulses at different points in the amplifier for two frequencies. In both cases, the blue waveform represents the input video signal, which has a DC skew of about 1.5 V and a pulse amplitude of about 250 mV.

The green and purple waveforms represent amplified pulses in neighboring stages. As can be seen, the circuit amplifies and does not distort the lower-frequency pulses. A greater distortion of the pulses can be observed at 100 kHz. Here we see not only a change in the slope of the pulse slopes but also in its levels. At this frequency, the pulses are still measurable by the AD converter. The observed distortion of the amplified pulses depends on the parameters of the used operational amplifiers. Two of them have the most influence: slew rate (SR) and gain bandwidth product (GBP). In the investigated circuit, popular amplifiers with a SR equal to 6.5 V/s and a GBP equal to 10 MHz were used. The wide frequency range of the amplification path is important because changing the clock pulse frequency allows controlling the integration time in the MEMS sensor. This time is an additional factor in adjusting the signal gain.

The DC value at the output of the MEMS sensor is about a single V. Such a high level does not require amplification but only filtering. The LP filter circuit used in the research is shown in Figure 17.

The cut-off frequency of the low-pass filter (Figure 18) was approximately 7 Hz. In the driver circuit, active filters were avoided due to the stability of the operational amplifiers in these filter circuits. 

The output signal from the AD converter is sent to the FPGA using the SPI bus. The FPGA module produces all the control signals for the MEMS spectrometer and the AD converter. In the presented system, a 10-bit chip [29] with a conversion rate of more than 200 ksps is sufficient as the AD converter. The details of the structure of the control components in the FPGA are not given in this paper, as this was a secondary issue in this case. The control module in the FPGA was developed using Intel’s PLD CAD software Quartus II Prime Light Edition [30]. As an example of the developed software, a top-level file view of one module is included (Figure 19). This is the module that generates the basic control signals for the MEMS sensor so that the MEMS spectrometer can be started up and its electrical signals measured.

The module includes the phase lock loop Petla, which divides the system frequency of the FPGA module to the user-required frequency of the MEMS sensor’s clock signal. The second module, Sterow, generates control signals for the MEMS spectrometer. It was developed using the finite state machine technique in the VHDL language. In reality, the complete FPGA software is more complex, as it also handles the processes of the analog to digital converter, digital processing of the measured signals, including scaling and averaging, and the main process that coordinates the component modules. Due to the breadth of the issue, the complete FPGA software is not discussed in the paper and will be the subject of another comprehensive article.

## 4. Conclusions

The aim of this study was to present an original, proprietary electronic driver to operate the MEMS optical sensor and condition the output signal. In the described case, the object of study was a miniature spectrometer operating in the visible radiation range. This type of optical signal-to-electrical signal converter still causes a lot of doubt and confusion among users. This is often due to the peculiarities of the technology and a misunderstanding of the requirements for the sensors. This is a technology in a growth phase and market solutions of similar spectrometers for other radiation ranges are expected in the near future. 

The author’s own contribution was the development of the original sensor control circuitry. To this end, the following were developed: the power supply circuits for the sensor, the amplification and filtering circuits, the method of converting the output signal into digital form, and the possibility of digital control from the FPGA level was confirmed. 

The MEMS spectrometer itself is not a standalone system, so once the power supply is connected, it will not operate autonomously. It needs the accompanying electronic circuitry for normal operation, which was the subject of this work. So, it is not possible to compare the characteristics of the sensor before and after the addition of the control circuit. The characteristics of the sensor are determined by the manufacturer in the technological process, so they cannot be improved afterward. An external circuit can only degrade the sensor characteristics. The author’s aim was to develop a control circuit that would not worsen the factory measurement characteristics. Thus, the amplifier was developed, which transmits and amplifies the output pulses over the entire operating frequency range of the sensor. This amplifier has a specific, proprietary ripple-curry architecture, in order to maintain a wide frequency response at low gain. At the same time, this amplifier-filter circuit implements a structured automatic gain control function, without the need for additional gain-level switching elements. Furthermore, the developed filter circuit [31] effectively separates the output DC component from the measured photoelectric signal.

Comparing the parameters of the MEMS sensor under study with other optical sensors would not add anything new to the scope of work and would unnecessarily lengthen the article. The subject of the paper was a sensor driver for a specific MEMS technology and not a comparative analysis of available MEMS sensors. Any other MEMS sensor chip will have different output signals and their parameters and a different way of handling them. Thus, the developed circuit is useful in the design of a greenhouse lamp simulator, and it is dedicated to the technology used in Hamamatsu’s MEMS C1266MA. In addition, the author of the paper is not familiar with MEMS spectrometers with serial output and amplitude modulation of the output signal, as in the studied chip. In the opinion of this work’s author, the selected and developed measurement path is predisposed to applications in flux stabilization systems. It is fast, mobile, and the parameters are sufficient for this type of application. The layout of the work is typical for reporting application research, so it deviates from the scheme commonly used in basic research.

## Figures and Tables

**Figure 1 sensors-23-02916-f001:**
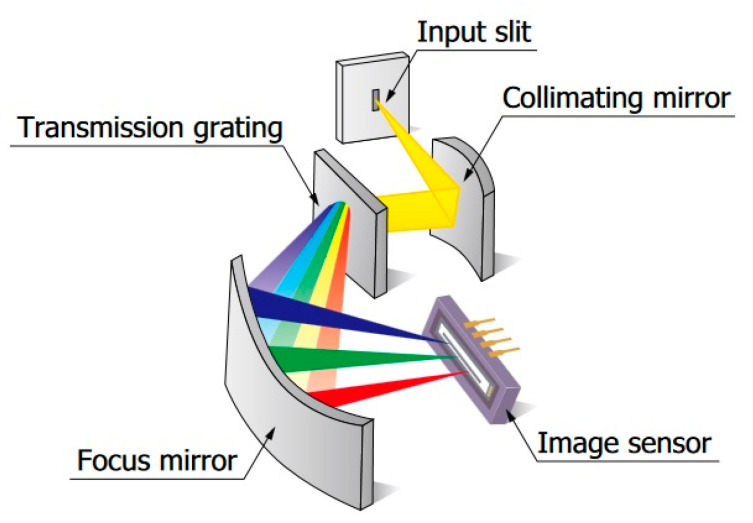
Sample architecture of an integrated monochromator in the mini-spectrometer [10].

**Figure 2 sensors-23-02916-f002:**
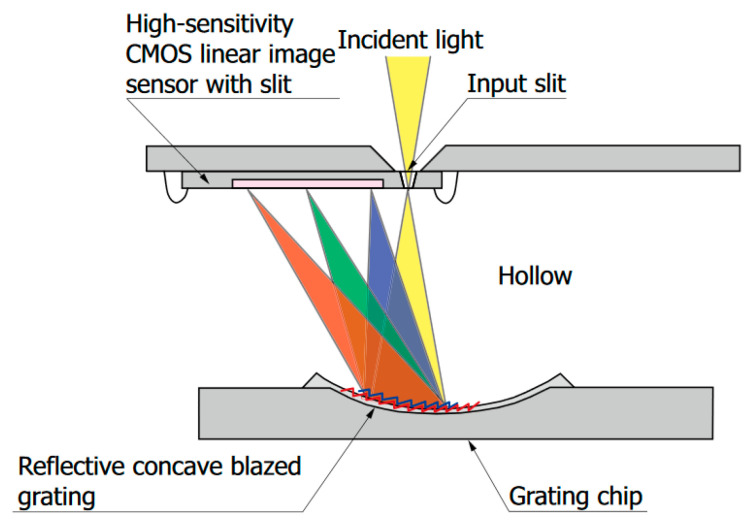
The internal structure of the MEMS spectrometer [23].

**Figure 3 sensors-23-02916-f003:**
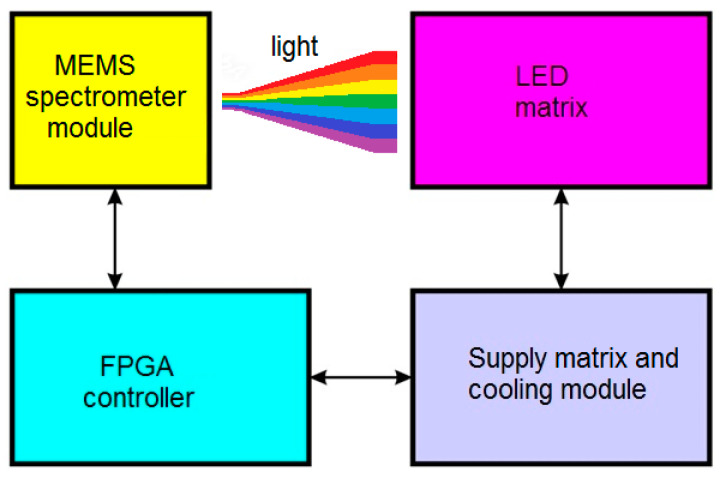
Block diagram of horticulture lamp hardware simulator.

**Figure 4 sensors-23-02916-f004:**
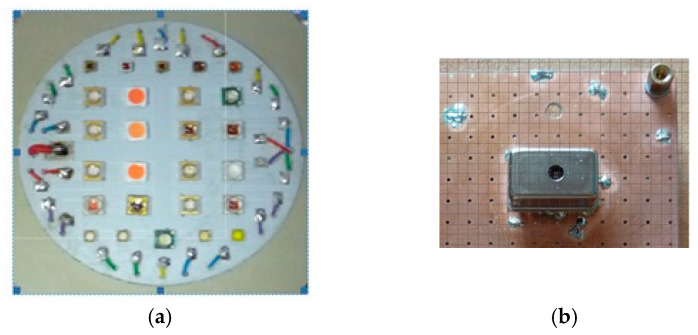
The view selected components of the hardware simulator: (**a**) the LED matrix on an aluminum substrate; (**b**) the MEMS sensor on the bottom side of the board MEMS spectrometer module.

**Figure 5 sensors-23-02916-f005:**
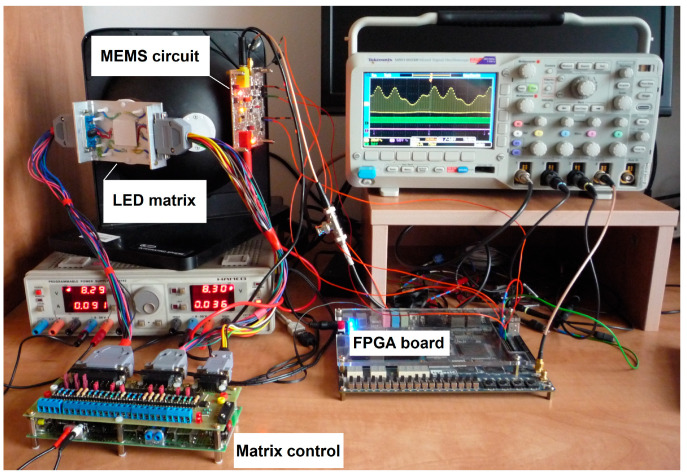
Measurement stand of prototype MEMS driver.

**Figure 6 sensors-23-02916-f006:**
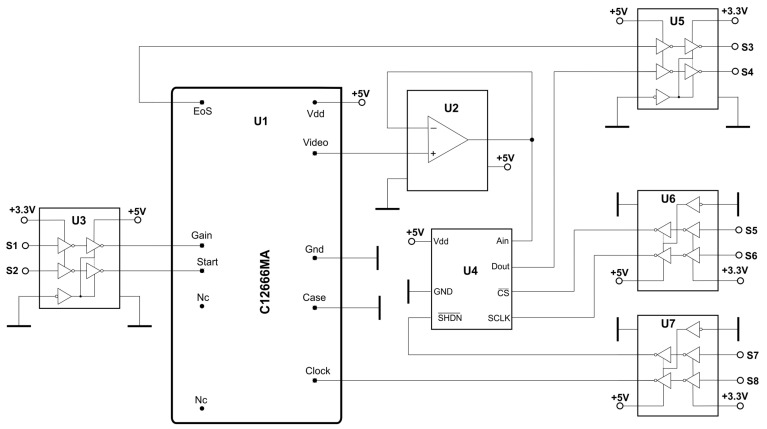
Default circuit of the C12666MA sensor.

**Figure 7 sensors-23-02916-f007:**
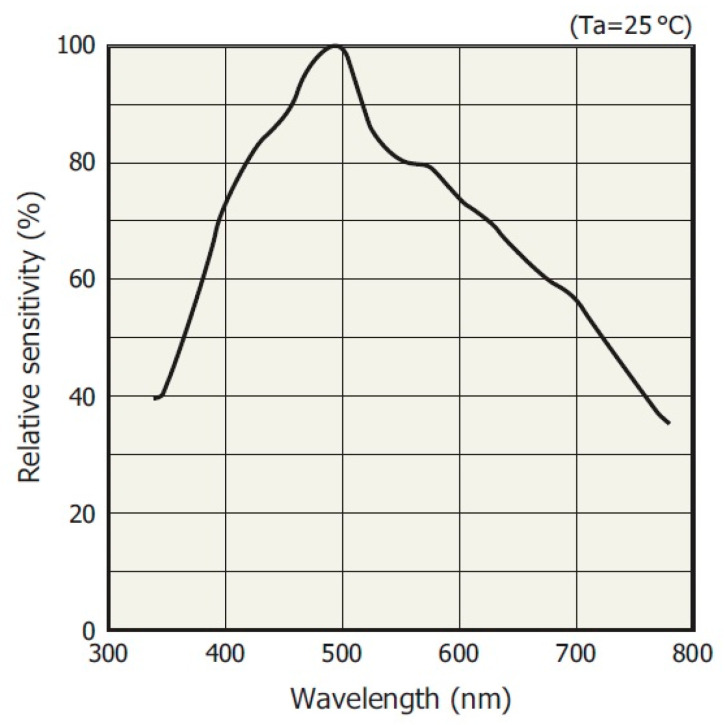
Typical spectral response of the MEMS sensor [23].

**Figure 8 sensors-23-02916-f008:**
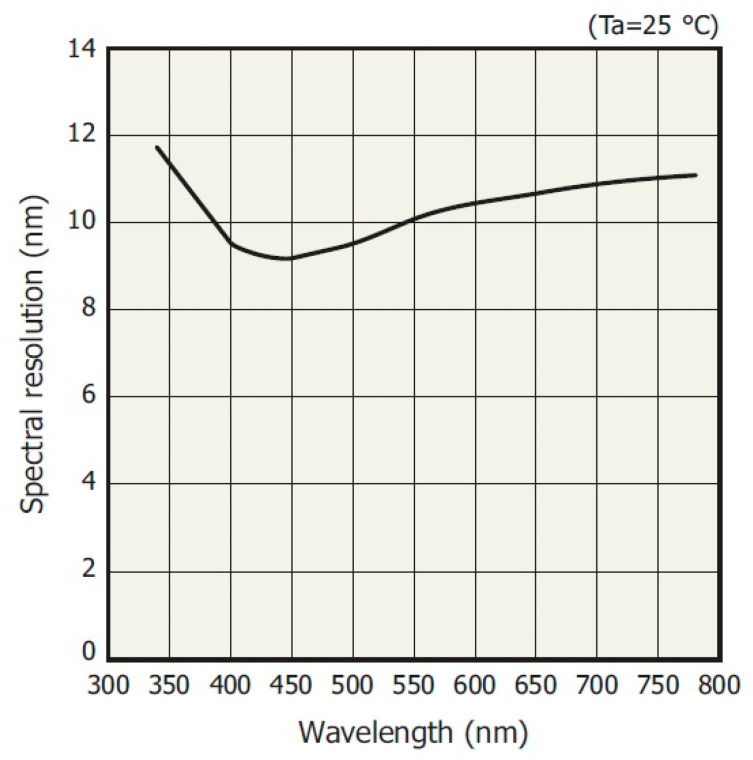
Typical spectral resolution vs. wavelength of the MEMS sensor [23].

**Figure 9 sensors-23-02916-f009:**
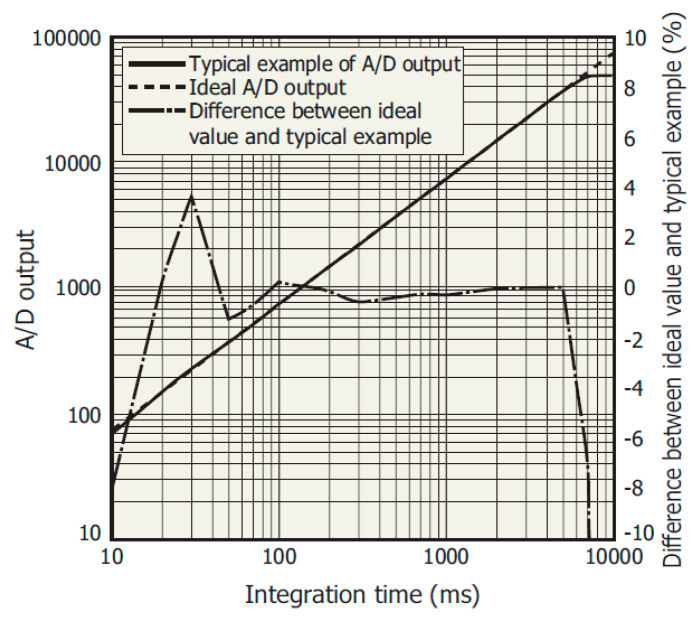
MEMS C12666CA linearity [23].

**Figure 10 sensors-23-02916-f010:**
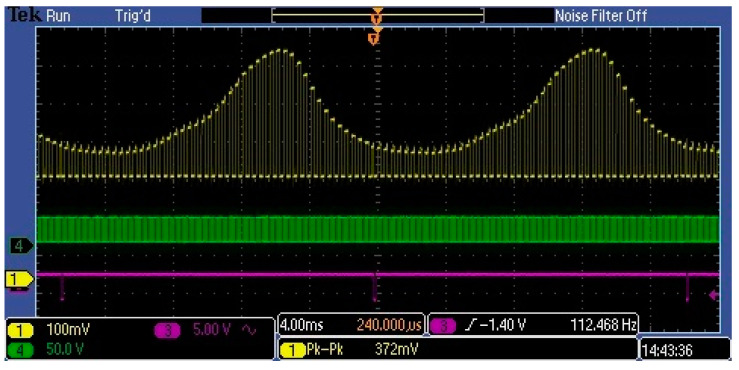
Measured waveforms of the MEMS spectrometer in the circuit shown in Figure 6: the yellow shows the Video signal, the green is the clock, and the purple is the end of scan C12666MA.

**Figure 11 sensors-23-02916-f011:**
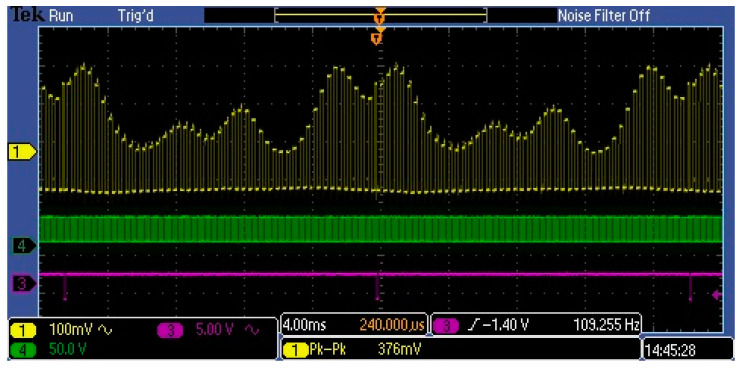
Measured MEMS waveforms when the *video* contains only the variable component.

**Figure 12 sensors-23-02916-f012:**
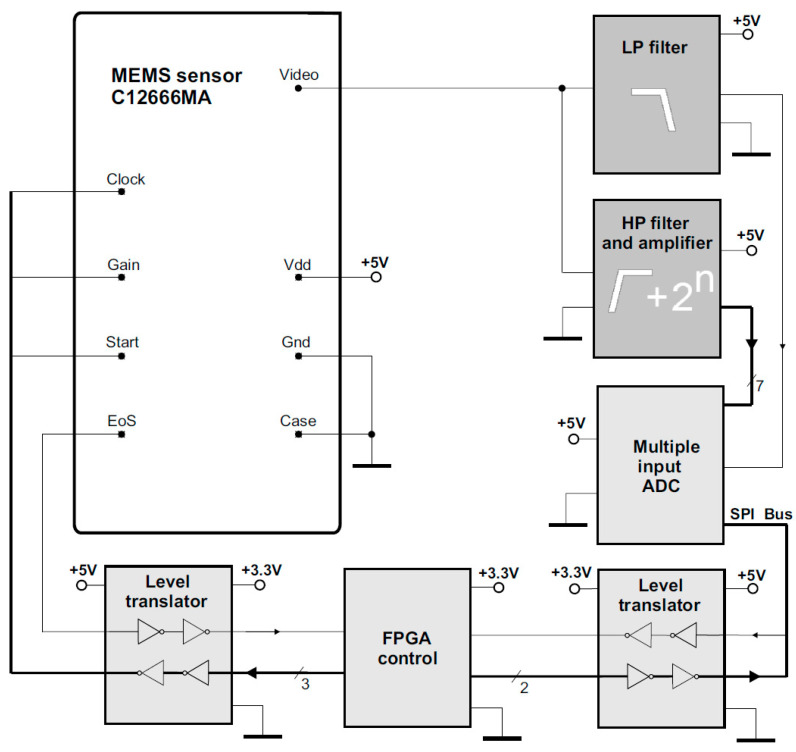
Block diagram of the new MEMS sensor circuit concept.

**Figure 13 sensors-23-02916-f013:**
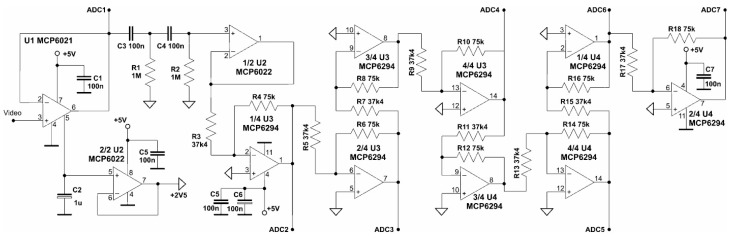
The tested video pulse conditioning circuit.

**Figure 14 sensors-23-02916-f014:**
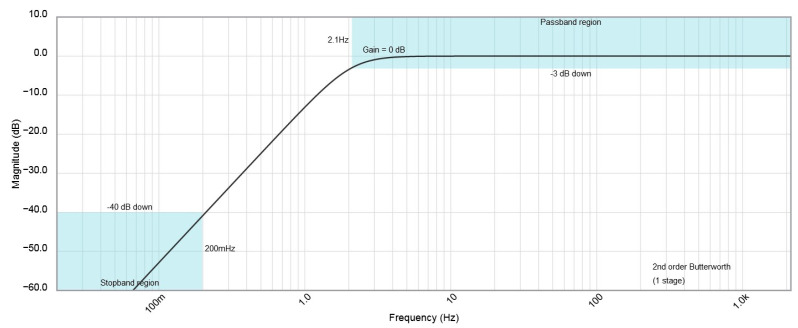
Simulation results of high-pass filter amplitude characteristics.

**Figure 15 sensors-23-02916-f015:**
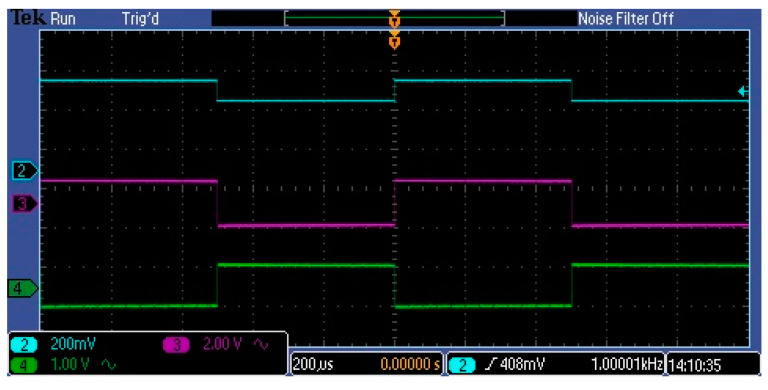
Amplified pulse waveforms at 1 kHz.

**Figure 16 sensors-23-02916-f016:**
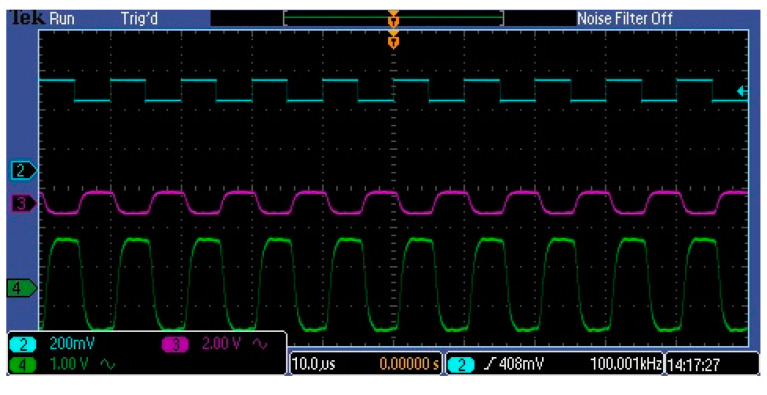
Amplified pulse waveforms at 100 kHz.

**Figure 17 sensors-23-02916-f017:**
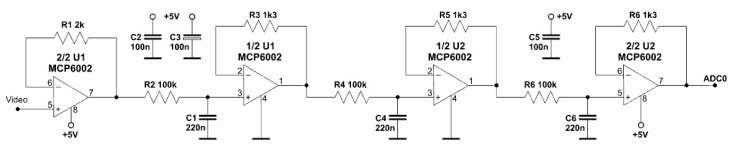
Low-pass filter schematic used in the measurement circuit.

**Figure 18 sensors-23-02916-f018:**
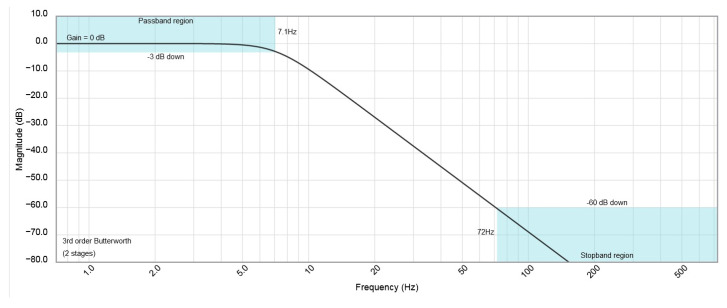
Simulation results of low-pass filter amplitude characteristics.

**Figure 19 sensors-23-02916-f019:**
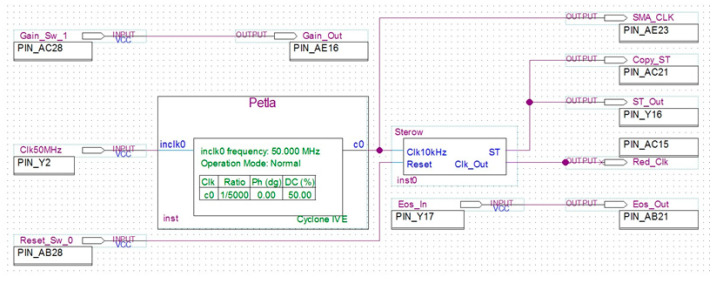
Simulation results of low-pass filter amplitude characteristics.

## Data Availability

The data is not publicly available because of continuing research work on the larger system, and the problem described in the article is the selected issue of the system being developed.

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
