# Peer review of "Micro-Electro-Mechanical Systems in Light Stabilization"

_sensors, 2023, doi:10.3390/s23062916_

Round 1

Reviewer 1 Report

The paper describes a micro electromechanical system for stabilizing the luminous flux from an LED lamp. However, the author did not clearly show what is the novelty of this device in comparison with analogues. Variants of already known MEMS spectrometers [10], [19] are analyzed. A matrix of 25 LEDs with cooling is taken from [20-22], and the LED current control algorithm is taken from [23]. The old sensor control circuit is shown in Fig. 6, and the new one in Fig. 9. An analogue-to-digital converter and high and low pass filters have been added to the new circuit. This made it possible to process rectangular pulses in a wide range from 200Hz to 200kHz. The proposed scheme allows amplifying the signal from the sensor up to 120 dB. In addition to these numbers, the paper does not provide other characteristics of the sensor: how many channels the spectrometer has, what is the resolution of the spectrometer, what is the range of measured wavelengths, what is the noise level of the output signal, and others. Therefore, I recommend publishing this work only after the author has significantly supplemented it with a comparison with known similar MEMS spectrometers and sensors and has given all the characteristics of the considered spectrometer before and after improvement.

Author Response

Dear Sir or Madam,

Thank you for your review.

My work is not about basic research, but about application research.

The subject of the research was the environment of the MEMS spectrometer, that is, the driver and amplifier that support it.

I agree with the review in terms of completing the parameters of the spectrometer, although when writing I was suggested by the volume of 10 pages of the communication and omitted some characteristics.

In this regard, I am able to complete the work.

To meet the recommendation to improve the work in some aspects is impossible for technical reasons, such as improving the parameters of the sensor, as they are predetermined by the factory, an external system can only make them worse. The purpose of the work was to use a MEMS spectrometer to measure light in an integrating sphere, not to improve its parameters.

I apologize for the misunderstood wording, which resulted in a misunderstanding of the role and scope of [20-22] and [23]. This literature is not related to the LED array or FPGA software. Items [20-22] are the results of basic and biological research, the light that plants need and not the design of the LED array. And item [23] is a description of the hardware board with FPGA and not the complete software that you need to write yourself. I send items [21-22], [20] can not send because it is a commercial standard, [23] I send the header because the file is 12 MB.

I am not familiar with similar market designs of MEMS spectrometers in this spectral range, so I have nothing to compare with. The world leader in this class of sensors, Hamamatsu, has a similar sensor, but older and with slightly lower performance, so a comparison with it is not substantive. In my application, it is not possible to use other types of spectrometers, such as compact, due to the miniaturization of the whole system. From my reconnaissance, it seems that a new MEMS sensor from Hamamatsu will be on the market this year, but for the UV range.

I cannot meet the expectation of comparing Hamatsu's MEMS C12666MA with other MEMS sensors or the parameters of the considered spectrometer after improvement, because I do not improve it only operate it.

Comparing the parameters of the MEMS sensor under study with other optical sensors would not bring anything new to the scope of work and would unnecessarily lengthen the article. The subject of the paper was a sensor controller for a specific MEMS technology. Any other MEMS sensor chip will have different output signals and their parameters and a different way of handling them. Thus, the developed circuit is useful for the greenhouse lamp simulator project and is dedicated to the technology used in Hamamatsu's MEMS C1266MA chip. In my opinion, the selected and developed measurement path is predisposed to applications in flux stabilization systems. It is fast, mobile, and its parameters are sufficient for such applications.

I am sending an additional file introducing the scope of the project, parts of which are covered in my article, and auxiliary files.

With Best Regards,

Marian Gilewski

Reviewer 2 Report

The author composed several chips to stabilize LED light. However, the MEMS chip is not developed by the author. And there are no control algorithms or control laws found in stabilization.  It is hard to found innovation and contribution in this work. 

Author Response

Dear Sir or Madam,

I thank you very much for your review.
My work is about applied research and not fundamental research. Hence, its nature is 'technical' rather than analytical. The motivation was the unsatisfactory state of the art and application research in the field of greenhouse lamps. I am familiar with the global market for greenhouse lamps and I am also a reviewer of about 100 application projects, including those related to horticulture technology. I am familiar with the unsatisfactory results of application research in my country, where the results in terms of spectral matching of lamps to the needs of plants is unsatisfactory. 
I cannot disclose the results of application research in the paper due to the obligation to protect the information of industrial research results. Such information is not published, but protected by company secrets or patents.

Hence, the concept of developing a reference light source or, rather, a hardware simulator of different lamps based on an integrating sphere (Figure 5) was born. A tool that could help engineers at the design stage. Such an adaptive system must work with feedback to stabilise its parameters, e.g. related to the ageing of the components.
It must include a miniature spectrometer for ongoing measurement of luminous flux. The best market solution, to my knowledge, is a MEMS sensor. There is a circuit that requires specific handling and the catalogue data does not show this. Hence, by incorrectly controlling this module, results with large measurement errors can be obtained. For this reason, some metrologists look critically at this module, even writing badly in the scientific forum. This is a future-oriented technology and the current year will bring market developments for the UV range. I wanted to reduce the negative attitude by writing this article. For the overall project, I developed my own original MEMS spectrometer control system, which is the subject of my article. Due to the volume of about 10 pages of Communication, I have concentrated only on describing the circuits involved in MEMS operation. In fact, I did not describe in detail the design process, simulation and only included the most relevant issues and example measurement results.
Hence, it was my aim to provide a new tool and show the method of using it. I will, of course, improve the article to the extent possible, describing in more detail the methodology and background to the development and use of the MEMS spectrometer. Due to limited time and the need to take into account the recommendations of the other reviewers, a radical rearrangement of the paper is not possible. I ask for your indulgence in this regard, as the work concerns applied research.

With Best Regards,
Marian Gilewski

Reviewer 3 Report

This manuscript presented the author's electronic circuitry that supports the optical MEMS sensor.

1\The current title does not match the main work in this manuscript. 

2\Can you list some quantitative advances in the abstract? 

3\ What is the novelty or significant progress in such a proposed system?

4\The manuscript should be revised in a form of scientific paper instead of an engineering report. 

Author Response

Dear Sir or Madam,

Thank you very much for your review.

My work concerns applied research, not basic research. Hence, its nature is "technical" rather than analytical, and therefore it was not prepared in a rigid rigor for basic research, but in a descriptive mode. It seemed to me that the Publisher accepts a certain amount of freedom for the author in this regard.

The immediate motivation was the unsatisfactory state of greenhouse lamp technology and application research. I am familiar with the global greenhouse lamp market, and I am also a reviewer of about 100 application projects, including those related to horticultural technology.

I am also familiar with the unsatisfactory results of application research in my country, where the results in terms of spectral matching of lamps to the needs of plants are inadequate.

The details of the application research and its results I cannot disclose in the paper due to the obligation to protect information about the results of industrial research. Such information is not published, but protected by company secrets or patents.

Experiencing such a situation, the concept of developing a reference LED light source, or rather, a hardware simulator for various lamps, which is based on an integrating sphere (Figure 5), was developed. This would be a tool that could help engineers at the design stage. Such a stable adaptive system must work with feedback to stabilize its parameters, such as those related to aging components.

It must include a miniature spectrometer for ongoing flux measurement. The best market solution for miniature spectrometers, to my knowledge, is a MEMS sensor. This is a system that requires specific handling, and its catalog data does not show this. Hence, by improperly controlling this module, results with large measurement errors can be obtained.

For this reason, some metrologists look critically at this module, even writing badly in the scientific forum.

This is a future-oriented technology and the current year will bring market development in the field of MEMS UV.

In writing this article, I wanted to reduce negativity.

My article deals only with a part of the project for which I developed my own original MEMS spectrometer control system. Due to the volume of about 10 pages of the communication, I focused only on describing the circuits responsible for MEMS operation. In fact, I did not describe in detail the design or simulation process, and included only the most relevant issues and sample measurement results.

My goal was to give an applied presentation of the new tool and show the method of its use.

Of course, as the article improves, I will describe in more detail the methodology and subject background and how to use the MEMS spectrometer. Due to limited time and the need to take into account the recommendations of the other reviewers, a radical reorganization of the article is not possible.

I cannot change the title, especially since it has been accepted by the other reviewers.

I am able to provide quantitative parameters in the abstract.

I will try to emphasize more the elements of novelty and originality of the solution.

Unfortunately, the form, typical for the presentation of the results of applied research, does not quite fit the style of describing basic research. I ask for your understanding in this regard.

With Best Regards,

Marian Gilewski

Round 2

Reviewer 1 Report

The authors took into account all my comments and I recommend publishing this work.

Author Response

I thank you very much for your valuable opinions and comments.
My best regards,
Marian Gilewski

Reviewer 2 Report

Although the authors claimed thier work is technique. However, there must be significant contributions of academic innovention in research paper. In this paper, we can not found any sinificant contribution on control law and algorithm. There are no novel MEMS  techiniques in this paper. Therefore, I think this paper has no importance in this topic.

Author Response

There are two research works, defined by the European Commission, among others.
These are basic research and applied research.
These are equally important and equally valuable types of research.
Basic research is about learning new laws of physics and developing, among other things, new electronic components - based on these laws. I don't work in the field of application of basic research - hence the novelty in my work is a new application, a new product or application circuit. The development of a new topology is a contribution to science, is a contribution to application research.
My new layout and new concept is the layout in Figure 12, this is my contribution to the field.

I have not found a similar one in the literature and no one will find one.
In the rest of the paper, I describe the individual blocks whose solution is my original one, which is natural in documenting application work.
My development is also the application in Figure 19 - I didn't steal it from any CD [28], because there you can find datasheets of the FPGA prototype module and examples of creating projects for students but not ready-made solutions for controlling the MEMS spectrometer.
The purpose of my paper is the application development of a specific sensor and not a review of possible MEMS sensors, because there are thousands of them and their review contributes nothing to my solution.
The article is about a specific application and not a review of this class of sensors.
I only refer to optical MEMS spectrometers - I have not found others similar to the one I have applied and operate.
If there are any I would like to know about them, as the best proof is the example.
Nevertheless, in the version I am sending, I have added 4 literature items - which indicate other applications.
If there are objections then please give me a concrete hint:
- which relevant references I have omitted,
- which quoted sources to remove as non-significant,
- in which my research is inadequate,
- which method I should describe more.
It will be easier for me to improve my work.
Yes I wrote earlier I completed the literature items.
Thank you very much for your time.
With best regards,
Marian Gilewski

Reviewer 3 Report

My concerns are cleared. The manuscript could be accepted, although the novelty is not high.

Author Response

I, for one, thank you very much for your time and valuable comments.
With best regards,
Marian Gilewski